# The relationship between congenital heart disease and cancer in Swedish children: A population-based cohort study

**Christina-Evmorfia Kampitsi** *, **Hanna Mogensen**, **Maria Feychting**, **Giorgio Tettamanti**

Unit of Epidemiology, Institute of Environmental Medicine, Karolinska Institutet, Stockholm, Sweden

* christina.evmorfia.kampitsi@ki.se

## Abstract

### Background

Birth defects have been consistently associated with elevated childhood cancer risks; however, the relationship between congenital heart disease (CHD) and childhood cancer remains conflicting. Considering the increasing patient population with CHD after improvements in their life expectancies, insights into this relationship are particularly compelling. Thus, we aimed to determine the relationship between CHD and cancer in Swedish children.

### Methods and findings

All individuals registered in the Swedish Medical Birth Register (MBR) between 1973 and 2014 were included in this population–based cohort study ($n = 4,178,722$). Individuals with CHD ($n = 66,892$) were identified from the MBR and National Patient Register, whereas cancer diagnoses were retrieved from the Swedish Cancer Register. The relationship between CHD and childhood cancer (<20 years at diagnosis) was evaluated using Cox proportional hazards regression models. We observed increased risks of cancer overall, leukemia, lymphoma, and hepatoblastoma in children with CHD, but after adjustment for Down syndrome, only the increased lymphoma (hazard ratio (HR) = 1.64, 95% confidence interval (CI) 1.11 to 2.44) and hepatoblastoma (HR = 3.94, 95% CI 1.83 to 8.47) risk remained. However, when restricting to CHD diagnoses from the MBR only, i.e., those diagnosed around birth, the risk for childhood cancer overall (HR = 1.45, 95% CI 1.23 to 1.71) and leukemia (HR = 1.41, 95% CI 1.08 to 1.84) was more pronounced, even after controlling for Down syndrome. Finally, a substantially elevated lymphoma risk (HR = 8.13, 95% CI 4.06 to 16.30) was observed in children with complex CHD. Limitations of the study include the National Patient Register not being nationwide until 1987, in addition to the rareness of the conditions under study providing limited power for analyses on the rarer cancer subtypes.

### Conclusions

We found associations between CHD and childhood lymphomas and hepatoblastomas not explained by a diagnosis of Down syndrome. Stronger associations were observed in complex CHD.

**Data Availability Statement:** Anonymized personal data were obtained from national Swedish Registry holders after ethical approval and privacy assessment. According to Swedish laws and

regulations, personal sensitive data can only be made available for researchers who fulfill legal requirements for access to personal sensitive data. Eligible individuals can apply for the data from the National Board of Health and Welfare in Sweden (https://www.socialstyrelsen.se/statistik-och-data/bestalla-data-och-statistik/) and from Statistics Sweden (https://www.scb.se/vara-tjanster/bestall-data-och-statistik/).

**Funding:** MF was supported by grants from the Swedish Cancer Society (www.cancerfonden.se; grant reference number 2015/724) and the Swedish Research Council (www.vr.se; grant reference number 2017-02913). The funders had no role in study design, data collection and analysis, decision to publish, or preparation of the manuscript.

**Competing interests:** The authors have declared that no competing interests exist.

**Abbreviations:** CHD, congenital heart disease; CI, confidence interval; CNS, central nervous system; HR, hazard ratio; ICCC-3, International Classification of Childhood Cancer, third edition; ICD, International Classification of Diseases; LISA, longitudinal integrated database for health insurance and labor market studies; MBR, Medical Birth Register.

## Author summary

### Why was this study done?

- Birth defects have consistently been associated with an increased childhood cancer risk, although for congenital heart disease (CHD), the associations have been conflicting.

- Increasing the understanding of which birth defects predispose toward specific cancer types is key to carrying out targeted interventions for earlier cancer diagnosis.

- Children with CHD are living longer than ever before; therefore, insights into the relationship with cancer are particularly compelling.

### What did the researchers do and find?

- By combining high–quality Swedish health and population registers, 4,178,722 children born between 1973 and 2014 were included in this study—including 66,892 children with CHD.

- The findings showed that children born with CHD were at increased risk of lymphoma and hepatoblastoma, whereas an initially observed increased risk of leukemia was explained by Down syndrome.

- The risk of lymphoma was particularly elevated in children with complex CHD.

### What do these findings mean?

- These results suggest an association between CHD and childhood cancer and have the potential to facilitate early cancer detection by increasing awareness among physicians.

- Further research should focus on the potential causal mechanisms underlying these findings, including genetics and radiation from cardiac procedures.

## Introduction

Birth defects or congenital anomalies affect about 5% to 7% of newborns [1] and are the leading cause of infant mortality [2]. They also have a profound impact on live-born children, causing an array of long-term physical, physiologic, or developmental disabilities—including cancer [2,3]. Congenital heart disease (CHD) constitutes the most common birth defect and occurs in approximately 1% of all births [4]. In recent years, the prevalence of CHD has increased due to advances in pediatric care and surgical therapies [5]; thus, children with CHD are increasingly living into adulthood [4].

Several studies have explored the relationship between birth defects and childhood cancer risk, with current evidence pointing toward a higher cancer incidence in children with birth defects [3,6,7]. A recent systematic review noted an increased risk for overall childhood cancer,

as well as for certain specific birth defect and cancer subtype combinations [8]. A leading theory is that a common genetic abnormality impairing normal development is predisposing toward both birth defects and subsequent cancerous growth [8]. However, there is limited research regarding the relationship between CHD and cancer risk. Some studies suggest a potentially elevated cancer risk in both children and adults among patients with a diagnosis of CHD [4,7,9,10]. The suggested biological mechanisms underlying such a relationship include shared genetic mutations [11], radiation exposure from imaging and therapeutic cardiac procedures, lifestyle adaptations owning to the disease [10], and the association with cancer-linked syndromes such as Down syndrome [12,13]. In fact, in a recent register-based Danish study, the modestly elevated cancer risk observed among persons with CHD disappeared when excluding persons with Down syndrome [12].

The relationship between CHD and cancer risk, therefore, remains conflicting and poorly understood. Thus, our aim was to utilize the population-based Swedish health data registers to determine the relationship between CHD and cancer risk in Swedish children and adolescents and evaluate whether CHD complexity affects the association.

## Methods

This population-based cohort study is reported as per the Strengthening the Reporting of Observational Studies in Epidemiology (STROBE) guidelines (S1 STROBE Checklist) and was conducted according to a prospective analysis plan (S1 Analysis Plan). In addition to this plan, we evaluated the relationship between CHD complexity and childhood cancer risk, as well as 2 additional cancer subtypes: hepatoblastoma and neuroblastoma.

The Swedish Medical Birth Register (MBR), established in 1973 and comprising data on practically all deliveries in Sweden [14], was used to identify the study population: All individuals born in Sweden between 1973 and 2014 and registered in the MBR were eligible for inclusion ($n$ = 4,282,914; S1 Fig). The MBR records, among other information, health conditions detected during prenatal, delivery, and neonatal care, coded according to the International Classification of Diseases (ICD). To identify children with CHD not diagnosed at birth, we also linked the study population to the National Patient Register, by using the unique personal identity number [15]. The Patient Register, initiated in 1964, achieved universal coverage of inpatient care in Sweden in 1987; outpatient care is also covered since 2001 [16]. Individuals were considered as having CHD if they had been assigned at least 1 diagnostic code for CHD in either register ($n$ = 66,892). As potentially increased medical monitoring in relation to a cancer diagnosis might result in detecting mild forms of CHD that might have remained otherwise unnoticed, only individuals with a CHD diagnosis from the Patient Register at least 2 years before any cancer diagnosis were classified as having CHD—whereas all children diagnosed with CHD in the MBR, i.e., at birth, were classified as having CHD. As the criterion introduced for considering diagnoses from the Patient Register might lead to individuals who do not develop cancer having more time to be diagnosed with CHD, we further restricted the CHD diagnosis up to age 19 (a deviation from our original analysis plan). As CHD is a condition present from birth regardless of when it is diagnosed, individuals with a diagnosis from the Patient Register who fulfilled the above criteria were considered exposed from birth. Additionally, CHD was categorized into mild–moderate and severe based on the ICD codes reported in the MBR or the Patient Register, according to the categorization used by Collins and colleagues [4], adapted from Warnes and colleagues [17] (S1 Table). If several codes were reported for the same individual, the categorization was based on the most severe condition. The ICD codes used were often nonspecific, leading to an imperfect demarcation of CHD complexity. As this was most pronounced among mild and moderate CHD, we combined

these 2 categories. Finally, when the registered ICD code was general enough that it could apply to any category, the condition was categorized as mild–moderate.

The study population was further linked to the Swedish Cancer Register to retrieve cancer diagnoses. The Cancer Register was established in 1958 and covers the whole population; it is considered to have high quality and completeness and underreporting is generally low [18]. Cancers occurring before age 20 were considered as childhood and adolescent cancer, and the ICD versions used by the Cancer Register were categorized by the authors according to the International Classification of Childhood Cancer, third edition (ICCC-3) [19]. We included all childhood cancers, with separate analyses of the most common cancer types: central nervous system (CNS) tumors, leukemia, and lymphoma. In addition, based on findings in previous studies [7], we also conducted separate analyses of hepatoblastoma and neuroblastoma.

Children enrolled in this population-based cohort study were followed from birth until the first of the following events: cancer diagnosis, death, emigration, 20th birthday, or end of follow-up (December 2015). Information regarding emigrations was obtained from the Total Population Register [20], while date of death was retrieved from the Cause of Death Register [21].

Descriptive statistics were used to report baseline characteristics of the study cohort. Hazard ratios (HRs) for total childhood cancer and subtypes and corresponding 95% confidence intervals (CIs) comparing children with and without CHD were estimated using Cox proportional hazards regression with attained age as the underlying time scale. Analyses were first adjusted for sex, birth decade, parental age, parental education, and region of residence at birth. In a separate model, we additionally adjusted for genetic syndromes predisposing to nervous system tumors (neurocutaneous syndromes)—as neurofibromatosis type I, the most common among these syndromes, has been associated with higher than expected frequency of CHD [22]. The fully adjusted model also included Down syndrome. These analyses were repeated for CHD diagnosed at birth, as a proxy of complexity. Subsequently, we also investigated CHD complexity, based on the categorization denoted above, to investigate if associations were particularly pronounced among children with more severe CHD. Finally, separate sensitivity analyses were performed: first including maternal smoking in the models to investigate whether it might drive some of the associations [23,24] (data only available in the MBR from 1982 onwards), and second with different lag periods between the CHD and cancer diagnoses (1 day, 6 months, 1 year, and ±2 years). Finally, we restricted to the years when the Patient Register had achieved nationwide coverage of inpatient care (1987 to 2015) to allow for better detection of CHD not diagnosed at birth.

Information on potential confounders such as sex, birth decade (1973 to 1979, 1980 to 1989, 1990 to 1999, 2000 to 2009, 2010 to 2015), region of residence at birth (six regions), and parental age (categorized as <20, 20 to 29, 30 to 34, 35 to 39, >39 years) was retrieved from the MBR. Moreover, information on conditions such as Down syndrome or neurocutaneous syndromes was obtained from the MBR, the Patient Register, and the Cause of Death Register. Parental education (categorized as primary, secondary, postsecondary) was retrieved from the national censuses and the longitudinal integrated database for health insurance and labor market studies (LISA) [25]. Those with missing information on the main covariates (i.e., the covariates in model 1) were excluded from the analysis (n = 104,192; 2.4%, S2 Table). Data were prepared with SAS 9.4, whereas all analyses were performed with STATA 16.1. Ethical approval for the current study was granted by the Regional Ethical Review Board, Stockholm, Sweden.

## Results

This study included 4,178,722 children accumulating a total of 64,642,606 person-years of follow-up time (median follow-up: 15.1 years). Among them, 66,892 (1.6%) had a diagnosis of CHD: 33,818 had a CHD diagnosis in the MBR, and, additionally, 33,074 had a CHD

diagnosed only in the Patient Register. Among all individuals with a CHD diagnosis, 4,729 (7.0%) had severe CHD; most of the children with severe CHD ($n = 3,169$) were diagnosed at birth. Individuals with CHD were more likely to have Down syndrome compared to the general population (4.3% versus 0.1%; Table 1).

Initial adjusted analyses (Table 2) showed an increased risk of childhood cancer overall (HR 1.49; 95% CI, 1.30 to 1.70), leukemia (HR 2.22; 95% CI, 1.80 to 2.73), lymphoma (HR 1.59; 95% CI, 1.07 to 2.36), and hepatoblastoma (HR 3.76; 95% CI, 1.75 to 8.09). Results remained unchanged when adjusting for neurocutaneous syndromes. However, after also adjusting for Down syndrome, the associations for childhood cancer overall and for leukemia disappeared, whereas the increased risk for lymphoma (HR: 1.64; 95% CI, 1.11 to 2.44) and hepatoblastoma (HR: 3.94; 95% CI, 1.83 to 8.47) remained essentially unchanged. Sex-stratified analyses showed that the increased risk for lymphoma was primarily observed among boys (HR: 1.94; 95% CI, 1.21 to 3.09), whereas the increased hepatoblastoma risk was seen in both boys (HR: 3.87; 95% CI, 1.41 to 10.65) and girls (HR: 4.07; 95% CI, 1.26 to 13.11). Results were not affected by adjustment for maternal smoking during pregnancy in a subset of the population (S3 Table), or when restricting to the years when the Patient Register has nationwide coverage of inpatient care (S4 Table). Shorter lag periods between obtaining a CHD diagnosis from the Patient Register and any cancer diagnosis revealed more pronounced overall cancer and lymphoma risks, although the difference between the main analysis and the

**Table 1. Baseline characteristics of the study cohort by CHD status.**

|  | CHD-free | CHD |
|---|---|---|
|  | n (%) | n (%) |
| **Sex** |  |  |
| Male | 2,114,396 (51.4) | 33,256 (49.7) |
| Female | 1,997,434 (48.6) | 33,636 (50.3) |
| **Maternal age (years)** |  |  |
| ≤20 | 215,901 (5.3) | 2,994 (4.5) |
| 20–29 | 2,139,319 (52.0) | 31,911 (47.7) |
| 30–34 | 1,167,759 (28.4) | 19,966 (29.9) |
| 35–39 | 493,540 (12.0) | 9,729 (14.5) |
| ≥40 | 95,311 (2.3) | 2,292 (3.4) |
| **Paternal age (years)** |  |  |
| ≤20 | 60,942 (1.5) | 925 (1.4) |
| 20–29 | 1,579,981 (38.4) | 23,068 (34.5) |
| 30–34 | 1,309,548 (31.9) | 21,580 (32.2) |
| 35–39 | 748,449 (18.2) | 13,105 (19.6) |
| ≥40 | 412,910 (10.0) | 8,214 (12.3) |
| **Maternal education** |  |  |
| Primary | 517,859 (12.6) | 8,208 (12.3) |
| Secondary | 1,877,081 (45.6) | 30,047 (44.9) |
| Postsecondary | 1,716,890 (41.8) | 28,637 (42.8) |
| **Paternal education** |  |  |
| Primary | 762,826 (18.6) | 11,325 (16.9) |
| Secondary | 1,995,423 (48.5) | 33,222 (49.7) |
| Postsecondary | 1,353,581 (32.9) | 22,345 (33.4) |
| **Neurocutaneous syndromes** | 2,677 (0.1) | 123 (0.2) |
| **Down syndrome** | 2,392 (0.1) | 2,899 (4.3) |

CHD, congenital heart disease.

**Table 2. Adjusted HRs (95% CIs) of cancer in children with CHD.**

|  | CHD/No CHD no. of cases | UNADJUSTED HR (95% CI) | MODEL 1 HR (95% CI) | MODEL 2 HR (95% CI) | MODEL 3 HR (95% CI) |
|---|---|---|---|---|---|
| **Total cancers** | 221/10,690 | 1.49 (1.30–1.70) | 1.49 (1.30–1.70) | 1.42 (1.24–1.62) | 1.05 (0.91–1.22) |
| Males | 103/5,756 | 1.33 (1.10–1.62) | 1.33 (1.10–1.62) | 1.27 (1.05–1.55) | 0.96 (0.78–1.18) |
| Females | 118/4,934 | 1.66 (1.39–2.00) | 1.66 (1.38–1.99) | 1.58 (1.32–1.90) | 1.15 (0.94–1.41) |
| **CNS** | 34/2,866 | 0.85 (0.61–1.20) | 0.87 (0.62–1.22) | 0.75 (0.54–1.06) | 0.75 (0.53–1.05) |
| Males | 14/1,493 | 0.69 (0.41–1.17) | 0.71 (0.42–1.19) | 0.61 (0.36–1.03) | 0.59 (0.35–1.01) |
| Females | 20/1,373 | 1.02 (0.66–1.59) | 1.04 (0.67–1.62) | 0.91 (0.58–1.41) | 0.92 (0.59–1.44) |
| **Leukemia** | 93/2,874 | 2.25 (1.83–2.76) | 2.22 (1.80–2.73) | 2.21 (1.80–2.72) | 0.92 (0.72–1.18) |
| Males | 39/1,585 | 1.78 (1.30–2.45) | 1.75 (1.27–2.40) | 1.75 (1.27–2.41) | 0.77 (0.54–1.11) |
| Females | 54/1,289 | 2.78 (2.12–3.65) | 2.75 (2.10–3.62) | 2.74 (2.08–3.60) | 1.06 (0.76–1.48) |
| **Lymphoma** | 25/1,248 | 1.54 (1.04–2.29) | 1.59 (1.07–2.36) | 1.58 (1.06–2.35) | 1.64 (1.11–2.44) |
| Males | 18/777 | 1.82 (1.14–2.90) | 1.86 (1.16–2.97) | 1.86 (1.16–2.96) | 1.94 (1.21–3.09) |
| Females | 7/471 | 1.13 (0.54–2.38) | 1.15 (0.55–2.43) | 1.14 (0.54–2.41) | 1.19 (0.56–2.50) |
| **Hepatoblastoma** | 7/115 | 4.08 (1.90–8.75) | 3.76 (1.75–8.09) | 3.77 (1.75–8.10) | 3.94 (1.83–8.47) |
| Males | 4/68 | 4.11 (1.50–11.28) | 3.70 (1.35–10.18) | 3.71 (1.35–10.20) | 3.87 (1.41–10.65) |
| Females | 3/47 | 4.08 (1.27–13.12) | 3.88 (1.20–12.52) | 3.89 (1.21–12.54) | 4.07 (1.26–13.11) |
| **Neuroblastoma** | 8/393 | 1.34 (0.66–2.69) | 1.23 (0.61–2.47) | 1.22 (0.61–2.46) | 1.18 (0.58–2.41) |
| Males | 4/220 | 1.24 (0.46–3.35) | 1.15 (0.43–3.08) | 1.14 (0.42–3.06) | 1.03 (0.37–2.87) |
| Females | 4/173 | 1.45 (0.54–3.92) | 1.32 (0.49–3.56) | 1.32 (0.49–3.57) | 1.38 (0.51–3.72) |

**Model 1:** adjusted for birth decade, maternal/paternal age and education, region of residence at birth.

**Model 2:** adjusted for birth decade, maternal/paternal age and education, region of residence at birth, neurocutaneous syndromes.

**Model 3:** adjusted for birth decade, maternal/paternal age and education, region of residence at birth, neurocutaneous syndromes, Down syndrome.

CHD, congenital heart disease; CI, confidence interval; CNS, central nervous system; HR, hazard ratio.

analysis with a 1-year lag period was small (S5 Table). When considering CHD diagnoses that occurred in the period ±2 years from the cancer diagnosis, a small increased risk of leukemia was also observed (HR 1.29: 95% CI 1.03 to 1.61).

When restricting analyses to children diagnosed with CHD at birth, the risks for cancer overall (HR 1.45; 95% CI, 1.23 to 1.71) and leukemia (HR 1.41; 95% CI, 1.08 to 1.84) were more pronounced and less affected by Down syndrome (S6 Table). However, in this analysis, no statistically significant association was observed between CHD and lymphoma risk (HR 1.15; 95% CI, 0.62 to 2.14). When examining the impact of CHD complexity (mild–moderate or severe) on cancer risk, there was a substantially increased risk of lymphoma among children with severe CHD (HR 8.13; 95% CI, 4.06 to 16.30), compared to the CHD-free population (Table 3).

Analyses regarding the association between CHD and subtypes of leukemia, lymphoma, and CNS tumors are presented in Table 4. The increased risk of lymphomas seemed to predominantly stem from non-Hodgkin's lymphoma (HR 2.04; 95% CI, 1.01 to 4.13), which, as in overall lymphoma, was more pronounced in boys (HR 2.27; 95% CI, 1.01 to 4.13). As with previous results, Down syndrome accounted for the risk of leukemia but not lymphoma. There was a substantial coexistence of CHD and Down syndrome in cases of leukemia: 50 of 93 children with CHD who developed leukemia also had Down syndrome (lymphoid leukemia: 24/57, acute myeloid leukemia: 22/27).

## Discussion

In this population-based cohort study, we found an increased risk of lymphoma, stemming from non-Hodgkin's lymphoma, and hepatoblastoma among children with CHD—while the

**Table 3. Adjusted HRs (95% CIs) of cancer in children by CHD complexity.**

|  | No. of cases | MODEL 2 HR (95% CI) | MODEL 3 HR (95% CI) |
|---|---|---|---|
| **Total cancers** |  |  |  |
| No CHD | 10,690 | ref | ref |
| Mild–moderate CHD | 203 | 1.39 (1.21–1.60) | 1.04 (0.89–1.20) |
| Severe CHD | 18 | 1.84 (1.16–2.92) | 1.29 (0.81–2.06) |
| **CNS** |  |  |  |
| No CHD | 2,866 | ref | ref |
| Mild–moderate CHD | 31 | 0.74 (0.52–1.05) | 0.73 (0.51–1.04) |
| Severe CHD | 3 | 1.01 (0.33–3.13) | 1.00 (0.32–3.11) |
| **Leukemia** |  |  |  |
| No CHD | 2,874 | ref | ref |
| Mild–moderate CHD | 90 | 2.28 (1.85–2.81) | 0.96 (0.75–1.23) |
| Severe CHD | 3 | 1.20 (0.39–3.71) | 0.42 (0.13–1.32) |
| **Lymphoma** |  |  |  |
| No CHD | 1,248 | ref | ref |
| Mild–moderate CHD | 17 | 1.15 (0.71–1.85) | 1.19 (0.74–1.93) |
| Severe CHD | 8 | 7.73 (3.85–15.49) | 8.13 (4.06–16.30) |

**Model 2:** adjusted for birth decade, maternal/paternal age and education, region of residence at birth, neurocutaneous syndromes.

**Model 3:** adjusted for birth decade, maternal/paternal age and education, region of residence at birth, neurocutaneous syndromes, Down syndrome.

CHD, congenital heart disease; CI, confidence intervals; CNS, central nervous system; HR, hazard ratio.

effect of CHD on the risk of all childhood cancers combined and childhood leukemia, particularly acute myeloid leukemia, was explained by Down syndrome. However, for children diagnosed with CHD at birth, Down syndrome did not entirely account for the increased total cancer and leukemia risks observed. Attempting to explain this finding, we hypothesized that one key difference between those diagnosed at birth and those diagnosed later in life might be the severity of the underlying condition. Indeed, when examining CHD complexity, lymphoma risk was considerably increased in severe CHD.

Previous research regarding the relationship between CHD and cancer risk is inconsistent. In agreement with our findings, a population-based Danish cohort study including 15,905 CHD patients found no effect of CHD on overall cancer risk after excluding Down syndrome patients [12], while a Californian population-based cohort study found elevated total childhood cancer and lymphoma risks in CHD patients (excluding chromosomal anomalies) [4]. Yet, contrary to our findings of increased risk only in childhood lymphomas and hepatoblastomas, a population-based cohort study in Taiwan [10] and a Swedish matched cohort study [26] reported increased risk of childhood cancer overall among individuals with CHD. However, they included a wide range of ages with no separate consideration of childhood cancers, no adjustment for Down syndrome, and a short or no lag period between the CHD and cancer diagnoses. As increasing use of primary healthcare among children with cancer has been found already 6 months before their diagnosis [27], not including a sufficient lag period might result in some CHD diagnoses being identified only because of the increased monitoring preceding a cancer diagnosis—leading to an overestimation of the association. In our data, childhood cancer and lymphoma risks were becoming increasingly pronounced as the lag period between the CHD and cancer diagnoses was getting shorter—to avoid overestimating the association, we opted for a 2-year lag period for CHD detected later in life (i.e., retrieved from the

**Table 4. Adjusted HRs (95% CIs) of leukemia, lymphoma, and CNS cancer subtypes in children with CHD.**

| | CHD/No CHD no. of cases | MODEL 1 HR (95% CI) | MODEL 2 HR (95% CI) | MODEL 3 HR (95% CI) |
|---|---|---|---|---|
| **Lymphoid leukemia** | 57/2,225 | 1.76 (1.35–2.29) | 1.76 (1.35–2.29) | 0.98 (0.73–1.32) |
| Males | 20/1,244 | 1.14 (0.73–1.77) | 1.14 (0.73–1.78) | 0.62 (0.38–1.01) |
| Females | 37/981 | 2.48 (1.79–3.45) | 2.48 (1.79–3.45) | 1.41 (0.96–2.07) |
| **Acute myeloid leukemia** | 27/410 | 4.49 (3.04–6.64) | 4.46 (3.02–6.59) | 0.74 (0.46–1.19) |
| Males | 13/208 | 4.50 (2.56–7.89) | 4.50 (2.57–7.90) | 0.96 (0.48–1.92) |
| Females | 14/202 | 4.49 (2.61–7.74) | 4.43 (2.57–7.63) | 0.57 (0.29–1.10) |
| **Hodgkin's lymphoma** | 7/611 | 0.94 (0.45–1.99) | 0.94 (0.45–1.98) | 0.98 (0.46–2.06) |
| Males | 5/323 | 1.30 (0.54–3.14) | 1.30 (0.54–3.14) | 1.35 (0.56–3.27) |
| Females | 2/288 | 0.56 (0.14–2.25) | 0.56 (0.14–2.24) | 0.58 (0.14–2.32) |
| **Non-Hodgkin's lymphoma** | 8/271 | 1.97 (0.97–3.98) | 1.96 (0.97–3.96) | 2.04 (1.01–4.13) |
| Males | 6/185 | 2.20 (0.97–4.96) | 2.17 (0.96–4.91) | 2.27 (1.01–5.13) |
| Females | 2/86 | 1.50 (0.37–6.11) | 1.50 (0.37–6.11) | 1.56 (0.38–6.36) |
| **Astrocytoma/other glioma** | 31/2,030 | 1.13 (0.79–1.61) | 0.94 (0.66–1.34) | 0.93 (0.65–1.34) |
| Males | 12/1,002 | 0.91 (0.52–1.61) | 0.74 (0.42–1.31) | 0.72 (0.41–1.29) |
| Females | 19/1,028 | 1.33 (0.84–2.09) | 1.12 (0.71–1.77) | 1.14 (0.72–1.80) |
| **Ependymoma** | 3/268 | 0.81 (0.26–2.53) | 0.78 (0.25–2.43) | 0.81 (0.26–2.53) |
| Males | 2/153 | 0.97 (0.24–3.93) | 0.96 (0.24–3.88) | 1.00 (0.25–4.03) |
| Females | 1/115 | 0.61 (0.09–4.38) | 0.57 (0.08–4.07) | 0.59 (0.08–4.23) |

**Model 1:** adjusted for birth decade, maternal/paternal age and education, region of residence at birth.

**Model 2:** adjusted for birth decade, maternal/paternal age and education, region of residence at birth, neurocutaneous syndromes.

**Model 3:** adjusted for birth decade, maternal/paternal age and education, region of residence at birth, neurocutaneous syndromes, Down syndrome.

CHD, congenital heart disease; CI, confidence interval; CNS, central nervous system; HR, hazard ratio.

Patient Register) in the main analyses. Finally, a large US register-based study reported a more than 2-fold increased total cancer risk among children with a CHD diagnosis; the highest risk was observed for hepatoblastomas and neuroblastomas [7]. The more pronounced risk increase compared to our study could be attributed either to genetic differences in the populations, or to the mix of active and passive registers utilized in the US study: The passive registers might have identified mainly children with severe CHD, diagnosed at birth, whereas including no lag period between the CHD and cancer diagnoses for the active registers could have overestimated the association.

While Down syndrome has not been consistently considered as a confounder in previous studies examining the relationship between CHD and cancer, our observation of an attenuated association when adjusting for Down syndrome, mainly observed for leukemia, is in agreement with previous research regarding the effect of Down syndrome on cancer risk. Down syndrome has been shown to be a risk factor of considerable strength for childhood leukemia, but not for solid tumors, such as lymphomas [13,28]. Indeed, when taking Down syndrome into account, the association between CHD and leukemia decreased considerably when considering CHD diagnosed at birth and was entirely explained away when considering CHD diagnoses also from the Patient Register. Results for lymphomas, on the other hand, did not vary greatly after the addition of Down syndrome as a covariate.

The potential mechanisms underlying the relationship between CHD and childhood cancer remain poorly understood. It has been suggested that teratogenesis and carcinogenesis may share a common foundation, with shared point mutations in specific genes, or developmental gene mutations occurring during early embryogenesis [11,29]. Radiation exposure from cardiac imaging and therapeutic procedures is also regarded as a potential noteworthy risk factor

for carcinogenesis in individuals with CHD [30]. Indeed, according to a recent large population-based study of adult patients in the Quebec CHD Database, higher exposure to low-dose ionizing radiation from cardiac procedures was associated with increased cancer risk, with indications of a dose–response relationship [31]. However, results from studies on low-dose ionizing radiation and cancer risk in children and young adults are conflicting [31] and have been criticized as likely to suffer from reverse causation or confounding by indication [32]. Additionally, there is some uncertainty regarding whether the induction period would be sufficient. While 3 to 5 years are considered sufficient for the manifestation of radiation-induced leukemia, solid tumors such as lymphomas typically will not appear for 10 to 15 years after radiation exposure (and commonly remain undiagnosed until adulthood) [30]. In our cohort, 8 of the 25 CHD patients with lymphoma had their CHD diagnosis <10 years before they were diagnosed with lymphoma. Finally, lifestyle adaptations owning to the disease (such as reduced physical activity) could potentially be another driving force behind the observed association between CHD and cancer risk; however, this might be more likely in adulthood, when lifestyle factors tend to play a larger role in carcinogenesis [10].

To our knowledge, this is the largest study to date on cancer risk among children with CHD in a European population. The register-based design allows for population-wide coverage and long-term follow-up, resulting in the largest possible number of cancer cases; additionally, it minimizes the potential for selection and recall bias. As such, we believe the data in this study to be representative of the Swedish population—and the findings generalizable to similar populations. The register link also allows for information on several relevant covariates, including the ability to identify patients with Down syndrome or neurocutaneous syndromes. Moreover, using both the MBR and Patient Register to retrieve CHD diagnoses allows for capturing children with CHD diagnosed later in life or not documented in the MBR.

One limitation of the current study is that both CHD and childhood cancer are rare: therefore, the statistical power for analyses of specific cancer subtypes was suboptimal for the rarer subtypes. Furthermore, in the beginning of the study, the Patient Register was not yet nationwide, although the coverage increased gradually during the first 14 years of the study period; it is therefore possible that some children whose CHD remained undetected at birth were not captured in the Patient Register either. If these children later developed cancer, they would have been considered as nonexposed, potentially leading to an underestimation of the association. However, when restricting to the 28 years of the study period when the Patient Register had nationwide coverage of inpatient care, results were similar. Therefore, we do not expect that this limitation has substantially impacted our estimates. Additionally, discounting diagnoses of CHD from the Patient Register less than 2 years before any cancer diagnosis might also lead to an underestimation of cancer risk among children with CHD. However, including them could have led to overestimating the association, as some might have remained undiagnosed if not for the increased monitoring in relation to the cancer diagnosis. Therefore, we have opted for a more conservative approach, but performed analyses with shorter lag periods as sensitivity analyses. We also performed an analysis in which we considered CHD diagnosis that occurred at least 2 years after the cancer, and in this analysis not only we observed stronger associations with overall childhood cancer risk, lymphomas, and hepatoblastomas, but we also found a small increased risk of leukemia. However, it is important to note that this sensitivity analysis could have overestimated some associations since it is likely that some children with cancer could have had an increased medical monitoring longer than 2 years after the cancer diagnosis. Further, the exclusion of individuals with missing information on the main covariates, albeit very small (2.6%), may potentially have introduced some selection bias. However, the distribution of both CHD and cancer was very similar in children with missing and complete information on covariates, rendering such an eventuality unlikely. Moreover, even

though access to some potential lifestyle-related covariates was limited by the information available in the registers, it bears mentioning that lifestyle adaptations due to the disease (such as reduced physical activity) are mediators of the association, rather than confounders. Similarly, comorbidities that might occur to a larger extent in severe CHD and lead to increased cancer surveillance would also be mediators of the association. Finally, the only other well-established risk factors for childhood cancer—prior chemotherapy and high-dose ionizing radiation [6]—cannot predate the occurrence of CHD and therefore cannot be considered as potential confounders.

This large population-based cohort study demonstrates an increased risk of non-Hodgkin's lymphoma and hepatoblastoma among children with CHD, independent of Down syndrome diagnosis. Strongest associations were found when taking CHD complexity into consideration. Physicians should be aware of this predisposition to cancer in individuals with CHD; this increased awareness may be key in facilitating early cancer diagnosis and improving subsequent management and outcome. Future research should investigate the causal mechanisms underlying these findings, such as genetics, or low-dose ionizing radiation from cardiac imaging and therapeutic procedures. In light of the increasing population with CHD, such insights can be particularly compelling.

## Supporting information

**S1 STROBE Checklist. Checklist of items that should be included in reports of *cohort studies*.**
(DOCX)

**S1 Analysis Plan. Prospective analysis plan.**
(DOCX)

**S1 Fig. Cohort flow chart.**
(PNG)

**S1 Table. List of ICD–10 codes and their representative lesions evaluated as severe.**
(DOCX)

**S2 Table. Distribution of exposure and main outcomes in the study population, according to completeness of data on the main covariates.** CHD, congenital heart disease; CNS, central nervous system.
(DOCX)

**S3 Table. Adjusted HRs (95% CIs) of cancer in children with CHD born from 1982 onwards.** CHD, congenital heart disease; CI, confidence interval; CNS, central nervous system; HR, hazard ratio.
(DOCX)

**S4 Table. Adjusted HRs (95% CIs) of cancer in children with CHD born from 1987 onwards.** CHD, congenital heart disease; CI, confidence interval; CNS, central nervous system; HR, hazard ratio.
(DOCX)

**S5 Table. Adjusted HRs (95% CIs) of cancer in children with CHD—different lag periods between CHD diagnosis from the Patient Register and cancer diagnosis.** CHD, congenital heart disease; CI, confidence interval; CNS, central nervous system; HR, hazard ratio.
(DOCX)

**S6 Table. Adjusted HRs (95% CIs) of cancer in children with CHD diagnosed from the MBR.** CHD, congenital heart disease; CI, confidence interval; CNS, central nervous system; HR, hazard ratio; MBR, Medical Birth Register.
(DOCX)

## Author Contributions

**Conceptualization:** Christina-Evmorfia Kampitsi, Maria Feychting, Giorgio Tettamanti.

**Data curation:** Christina-Evmorfia Kampitsi, Giorgio Tettamanti.

**Formal analysis:** Christina-Evmorfia Kampitsi.

**Funding acquisition:** Maria Feychting.

**Investigation:** Christina-Evmorfia Kampitsi, Maria Feychting, Giorgio Tettamanti.

**Methodology:** Christina-Evmorfia Kampitsi, Hanna Mogensen, Maria Feychting, Giorgio Tettamanti.

**Project administration:** Maria Feychting.

**Resources:** Maria Feychting.

**Supervision:** Maria Feychting, Giorgio Tettamanti.

**Writing – original draft:** Christina-Evmorfia Kampitsi.

**Writing – review & editing:** Christina-Evmorfia Kampitsi, Hanna Mogensen, Maria Feychting, Giorgio Tettamanti.

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
