## [Editor Report · Decision Letter 0]

5 Jul 2021

Dear Dr Kampitsi, 

Thank you for submitting your manuscript entitled "Cancer risk in children with congenital heart disease: results from a Swedish register-based study" for consideration by PLOS Medicine.

Your manuscript has now been evaluated by the PLOS Medicine editorial staff and I am writing to let you know that we would like to send your submission out for external peer review.

Please re-submit your manuscript within two working days, i.e. by Jul 07 2021 11:59PM.

Kind regards,

Callam Davidson

Associate Editor

PLOS Medicine

---

## [Decision Letter · Decision Letter 1]

7 Sep 2021

Dear Dr. Kampitsi,

Thank you very much for submitting your manuscript "Cancer risk in children with congenital heart disease: results from a Swedish register-based study" (PMEDICINE-D-21-02870R1) for consideration at PLOS Medicine. 

Your paper was evaluated by an associate editor and discussed among all the editors here. It was also discussed with an academic editor with relevant expertise, and sent to independent reviewers, including a statistical reviewer. The reviews are appended at the bottom of this email and any accompanying reviewer attachments can be seen via the link below:

[LINK]

In light of these reviews, I am afraid that we will not be able to accept the manuscript for publication in the journal in its current form, but we would like to consider a revised version that addresses the reviewers' and editors' comments. Obviously we cannot make any decision about publication until we have seen the revised manuscript and your response, and we plan to seek re-review by one or more of the reviewers. 

We hope to receive your revised manuscript by Sep 28 2021 11:59PM. Please email us (plosmedicine@plos.org) if you have any questions or concerns.

We look forward to receiving your revised manuscript. 

Sincerely,

Callam Davidson, 

Associate Editor

PLOS Medicine

plosmedicine.org

Please update your title to ‘The relationship between congenital heart disease and cancer in Swedish children: a population-based cohort study’.

Data Availability Statement (DAS) requires revision:

• Data availability statement: Please replace the word ‘secrecy’ with ‘privacy’.

• A study author cannot be the contact person for the data. Please see http://journals.plos.org/plosmedicine/s/data-availability and FAQs at http://journals.plos.org/plosmedicine/s/data-availability#loc-faqs-for-data-policy 

Please structure your abstract using the PLOS Medicine headings (Background, Methods and Findings, Conclusions).

• Please combine the Methods and Findings sections into one section, “Methods and findings”.

• In the last sentence of the Abstract Methods and Findings section, please describe the main limitation(s) of the study's methodology. 

Please remove the ‘Financial disclosures’ from the title page. This information is captured as metadata based on your responses to the submission form.

Please update your citations throughout: these should be in square brackets. 

Lines 159 and 161: PLOS does not permit "data not shown.” Please remove this claim, or if you are the owner of the data relevant to this claim, please provide the data in accordance with the PLOS data policy, and update your Data Availability Statement as needed.

Please refer to your study as a population-based cohort study in the Methods and Abstract.

Line 249: Please update to ‘To our knowledge, this is the largest study to date on cancer risk among children with CHD in a European population’.

Please discuss the risk of selection bias that could have been introduced by the exclusion of subjects with missing data.

Please ensure that the study is reported according to the STROBE guideline, and include the completed STROBE checklist as Supporting Information. Please add the following statement, or similar, to the Methods: "This study is reported as per the Strengthening the Reporting of Observational Studies in Epidemiology (STROBE) guideline (S1 Checklist)."

Did your study have a prospective protocol or analysis plan? Please state this (either way) early in the Methods section.

Comments from the reviewers:

Reviewer #1: The authors exanimated relationship between congenital heart disease (CHD) and childhood cancer risk. They aimed to determine the relationship between CHD and cancer in Swedish children. All individuals registered in the Swedish Medical Birth Register between 1973 and 2014 were included in the study. Individuals with CHD were identified from the MBR and National Patient Register, whereas cancer diagnoses were retrieved from the Swedish Cancer Register. The authors observed increased risks of childhood cancer overall, leukemia, lymphoma, and hepatoblastoma in children with CHD, but after adjustment for Down syndrome, only the increased lymphoma and hepatoblastoma risk remained. When restricting to CHD diagnoses from the MBR only, i.e., those diagnosed around birth, the risk for childhood cancer overall) and leukemia was more pronounced, even after controlling for Down syndrome. Moreover, they observed substantially elevated lymphoma risk in children with complex CHD. In conclusions: they found associations between CHD and childhood lymphomas and hepatoblastomas, not explained by a diagnosis of Down syndrome. They observed stronger associations in complex CHD.

The study was perfectly designed and conducted.

Reviewer #2: See attached document

Reviewer #3: I confine my remarks to statistical aspects of this paper. The general approach is fine, but i have some issues to resolve before I can recommend publication.

One big issue is that this is a population study -- that is, you have an entire population, not a sample. Many statisticians, including me, think that this makes inferential statistics odd. You don't have any inference to do, you have the whole population. Other statisticians say inference is still meaningful if you posit a sort of super-population (e,g, all Scandinavian children, all children anywhere ....) I'm not a big fan of this, but I wouldn't reject an article that did it. But it should be discussed. The authors (wisely, in my view) did not include p values but they did include confidence intervals. These have the same problem. I'd prefer leaving those out and putting in standard errors of the estimates.

On p. 6, lines 129-132, the authors categorize many continuous variables. This is nearly always a mistake. I wrote an article about this: https://medium.com/@peterflom/what-happens-when-we-categorize-an-independent-variable-in-regression-77d4c5862b6c but, briefly, leave the variables as is and use splines to investigate nonlinearity. 

Peter Flom

[LINK]

---

## [Decision Letter · Decision Letter 2]

8 Nov 2021

Dear Dr. Kampitsi,

Thank you very much for submitting your revised manuscript "The relationship between congenital heart disease and cancer in Swedish children: a population-based cohort study" (PMEDICINE-D-21-02870R2) for consideration at PLOS Medicine. 

The revised manuscript was seen again by the reviewers and their reviews are appended at the bottom of this email and any accompanying reviewer attachments can be seen via the link below:

[LINK]

In light of reviewer #2's remaining concerns, we will not be able to accept the manuscript for publication in the journal in its current form, but we would like to consider a revised version that addresses the reviewers' comments. We cannot make any decision about publication until we have seen the revised manuscript and your response, and we plan to seek re-review by reviewer #2. 

We hope to receive your revised manuscript by Nov 29 2021 11:59PM. Please email us (plosmedicine@plos.org) if you have any questions or concerns.

We look forward to receiving your revised manuscript. 

Sincerely,

Callam Davidson, 

PLOS Medicine

plosmedicine.org

Comments from the academic editor:

The immortal bias is still a concern. The authors may want to perform additional sensitivity analyses to assess how much of their results could potentially be explained by this bias.

Comments from the reviewers:

Reviewer #2: I received the authors' responses and found that they acknowledged the critical issues raised for the most part. However, some important issues are left unsolved. I, therefore, request a further revision on the following points:

1. I find the problem of exposure definition in this paper to be two-fold. Firstly, unlike acquired conditions, CHD is a condition that is present from birth, regardless of when this condition is 'diagnosed'. This study defines the exposure based on CHD diagnosis from patient registries in addition to birth registries. That could lead to bias if the diagnose date is considered the start date of CHD condition. In this study, a patient not 'diagnosed' with CHD at birth but later had a CHD diagnosis (in NPR) were classified 'exposed' since the diagnosis and "unexposed" between birth to the diagnosis. This is wrong. The patient should be classified as "exposed" since birth even though his/her CHD condition was diagnosed later. Secondly, in response to my previous comment 1, the authors stated that they classified a patient as 'unexposed' if their cancer diagnosis precedes the CHD diagnosis (in NPR). This approach raised the second concern of misclassification on exposure. Indeed, according to this classification, the unexposed patient would survive less than the exposed patients, giving an artificial advantage to them (exposed patients). In this regard, I would like to refer this paper [Lévesque et al. 2010] to understand the problem under discussion better. 

2. As requested, the authors provided evidence supporting their claim against the hypothesis that exposure to LDIR from cardiac procedures explains the CHD-Cancer association. However, the evidence (S5 Table. Adjusted HRs (95% CIs) of cancer in children with Congenital Heart Disease (CHD), stratified by age at cancer diagnosis) is not convincing. The authors claim that solid tumors such as lymphomas typically need 10 to 15 years after radiation exposure to manifest. However, there still seems to be some uncertainty regarding the latency between low-dose ionization radiation exposures and cancer incidence. For example, in a large population study in Australia (Mathews, J. D., et al. Cancer risk in 680 000 people exposed to computed tomography scans in childhood or adolescence: data linkage study of 11 million Australians. Bmj (2013) 346.) that assessed the association between CT scan and cancer in children and adolescents, it was reported that " At 1-4, 5-9, 10-14, and 15 or more years since first exposure, IRRs were 1.35 (1.25 to 1.45), 1.25 (1.17 to 1.34), 1.14 (1.06 to 1.22), and 1.24 (1.14 to 1.34), respectively." Thus, simply comparing the CHD-cancer association between groups with age at cancer diagnosis before and after 10 years of age could not be used as evidence for assessing the LDIR-cancer association among CHD children population.

There is a knowledge base that says otherwise and is well established based on large epidemiological studies (refs). As in this study cohort, however, stated by the authors in lines 280-281, "8 of the 25 CHD patients with lymphoma had their CHD diagnosis <10 years before they were diagnosed with lymphoma" (with longer latency). This goes back to my first point; how did they define their exposure. 

To summarize, the subgroup analysis by age at cancer diagnosis thus does not provide compelling evidence to claim that exposure to LDIR from cardiac procedures does not contribute the CHD-Cancer association. I would suggest that the authors refrain from including this sub-group analysis, which might question the internal validity, and to remove the unfounded statement on LDIR. 

Reference

Lévesque, L. E., Hanley, J. A., Kezouh, A., & Suissa, S. (2010). Problem of immortal time bias in cohort studies: example using statins for preventing progression of diabetes. BMJ: British Medical Journal, 340(7752), 907-911. http://www.jstor.org/stable/40701710

Reviewer #3: The authors have addressed my concerns and I now recommend publication

Peter Flom

[LINK]

---

## [Editor Report · Decision Letter 3]

22 Dec 2021

Dear Dr. Kampitsi,

Thank you very much for re-submitting your manuscript "The relationship between congenital heart disease and cancer in Swedish children: a population-based cohort study" (PMEDICINE-D-21-02870R3) for review by PLOS Medicine.

I have discussed the paper with my colleagues and the academic editor. I am pleased to say that provided the remaining editorial and production issues are dealt with we are planning to accept the paper for publication in the journal.

[LINK]

We look forward to receiving the revised manuscript by Dec 29 2021 11:59PM.   

Sincerely,

Callam Davidson, 

Associate Editor 

PLOS Medicine

plosmedicine.org

Requests from Editors:

Please update the bullet point at line 69 to ‘These results suggest an association between CHD and childhood cancer and have the potential to facilitate early cancer detection by increasing awareness amongst physicians.’

Please provide the unadjusted comparisons as well as the adjusted comparisons in Table 2.

---

## [Editor Report · Decision Letter 4]

5 Jan 2022

Dear Dr Kampitsi, 

On behalf of my colleagues and the Academic Editor, Dr Wei Zheng, I am pleased to inform you that we have agreed to publish your manuscript "The relationship between congenital heart disease and cancer in Swedish children: a population-based cohort study" (PMEDICINE-D-21-02870R4) in PLOS Medicine.

PRESS

Sincerely, 

Callam Davidson 

Associate Editor 

PLOS Medicine